# Mirror Image of Spontaneous Intracranial Hemorrhage

**DOI:** 10.3390/diagnostics14040357

**Published:** 2024-02-06

**Authors:** Xiao Dong, Yuanyuan Liu, Xuehong Chu, Erlan Yu, Xiaole Jia, Chuanjie Wu

**Affiliations:** Xuanwu Hospital, Capital Medical University, Beijing 100053, China; dongxiao000512@163.com (X.D.); yuanyuanliu496@163.com (Y.L.); cxhsxh@163.com (X.C.); yuerlan0824@163.com (E.Y.); 15933968308@163.com (X.J.)

**Keywords:** intracranial hemorrhage, bilateral basal ganglia, simultaneous, mirror, symmetrical, pathogenesis

## Abstract

In this paper, we reported the first case of mirrored spontaneous intracranial hemorrhage with almost identical hematoma morphological characteristics. This patient’s first symptom was loss of consciousness, without any local neurological symptoms. This clinical presentation fits well with the atypical computed tomography (CT) image showing bilateral hematomas, and indicates that the distribution of hypertensive vascular damage may be symmetric and that the degree of the bilateral lesions may be similar.

A 59-year-old female presented to the emergency department with a sudden loss of consciousness and complete quadriplegia. Her bilateral psychological reflection was weakened and the reflex of Babinski’s sign was positive. She had a history of poorly controlled and irregularly treated hypertension. Immediate non-contrast computed tomography (CT) revealed bilateral putaminal hemorrhages that were symmetrical (right side 10.7 mL, left side 11.37 mL). These two paired hemorrhages mirror each other in position and shape (Figure 1). The magnetic resonance imaging of the head and computed tomography angiography of the head and neck did not show aneurysms, vascular malformations, vascular amyloidosis, or tumors (Figure 2). This patient was considered a rare case of hypertensive spontaneous simultaneous bilateral intracranial hemorrhage (SSBICH) based on the following reasons: poor hypertension; the site of the hemorrhage, which is typical in hypertensive angiopathy; and no other etiology identified except for hypertension. The patient regained consciousness after two days of supported care in the intensive care unit. Nevertheless, there are residual psychiatric disorders, characterized by irritable personality alteration and behavioral abnormalities. These symptoms could be alleviated by sodium valproate, olanzapine, and chloral hydrate. Head CT scan one month later showed that the bilateral mirror hemorrhage had been almost completely absorbed before her discharge from the hospital. The patient’s modified Rankin Scale (mRS) score was 3 at three months.

Intracranial hemorrhage (ICH) is generally solitary and unilateral. With a prevalence ranging from 0.8% to 5.9% in all spontaneous ICHs [1,2,3,4], simultaneous multiple ICHs has been considered a rare entity. Simultaneous bilateral multiple ICHs account for nearly half of simultaneous multiple ICHs [4]. The most common site of SSBICH is the basal ganglia, and, among all subtypes, they have the highest incidence of unfavorable prognosis [5,6].

A widely used etiological classification for ICH is SMASH-U (structured vascular lesion, medication, amyloid angiopathy, systemic disease, hypertension, undetermined) [7]. The research of Wu et al. demonstrated that individuals with simultaneous multiple ICHs and those with a single ICH varied in the distribution of their etiologies [1]. There is a greater incidence of systemic disease and a smaller percentage of hypertensive angiopathy in multiple ICHs compared to single ICHs. However, Alhashim et al. stated that hypertension is still acknowledged as the most common underlying etiology of SSBICH [6]. The pathophysiological mechanism of hypertensive angiopathy is a lipohyalinotic change in the vessel wall, which typically develops in the small arteries that traverse deep brain tissue. Intracranial hemorrhage has been attributed to the rupture of perforating arteries or Charcot–Bouchard aneurysms induced by lipohyalinotic change [8].

There are two main underlying hypotheses for bilateral ICHs, proposed by Kabuto et al. [9]. First, bilateral microaneurysms or perforating arteries rupture simultaneously. Second, a blood pressure increase and hemodynamic changes after unilateral hemorrhage trigger the rupture of contralateral arteries. The interval between consecutive ruptures can range from a few seconds to several hours [10]. It is challenging to identify either of these mechanisms unless there is a distinct sequence of neurological symptoms on each side. Analyzing the location of hematomas in 41 patients with simultaneous bilateral hypertensive putaminal or thalamic hemorrhages, Kenichi et al. discovered that bilateral symmetric hemorrhages are more frequent [10]. Based on this phenomenon, a hypothesis that the distribution of lipohyalinotic change in perforating arteries is prone to be symmetric has been proposed. This symmetric distribution may make it easier for bilateral ruptures to happen simultaneously or over short intervals of time.

Our patient’s first symptom was the loss of consciousness without any unilateral local neurological symptoms. Her CT scan revealed bilateral hematomas with almost identical morphological characteristics. No prior research has documented such a mirror image of spontaneous cerebral bleeding. The short time lag and the mirror image are supportive evidence of our hypothesis that the distribution of lesions related to hypertension may be symmetric, which makes it more likely for bilateral ruptures to occur at the same time or over a very short amount of time. Additionally, it might suggest that both sides have comparable levels of hypertension vascular damage.

## Figures and Tables

**Figure 1 diagnostics-14-00357-f001:**
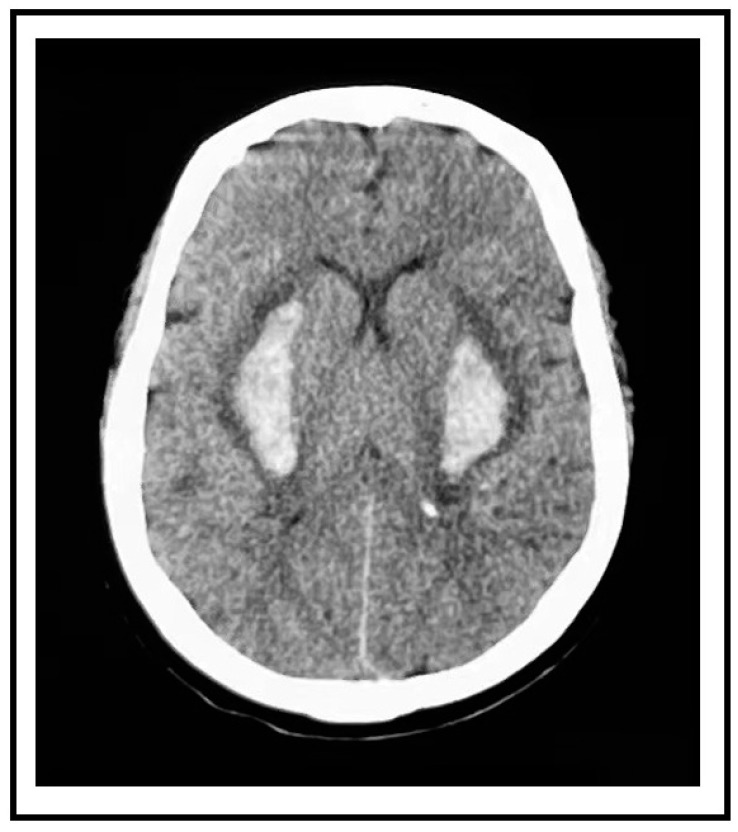
Immediate noncontrast head computed tomography showed a high-density image in the bilateral putamen.

**Figure 2 diagnostics-14-00357-f002:**
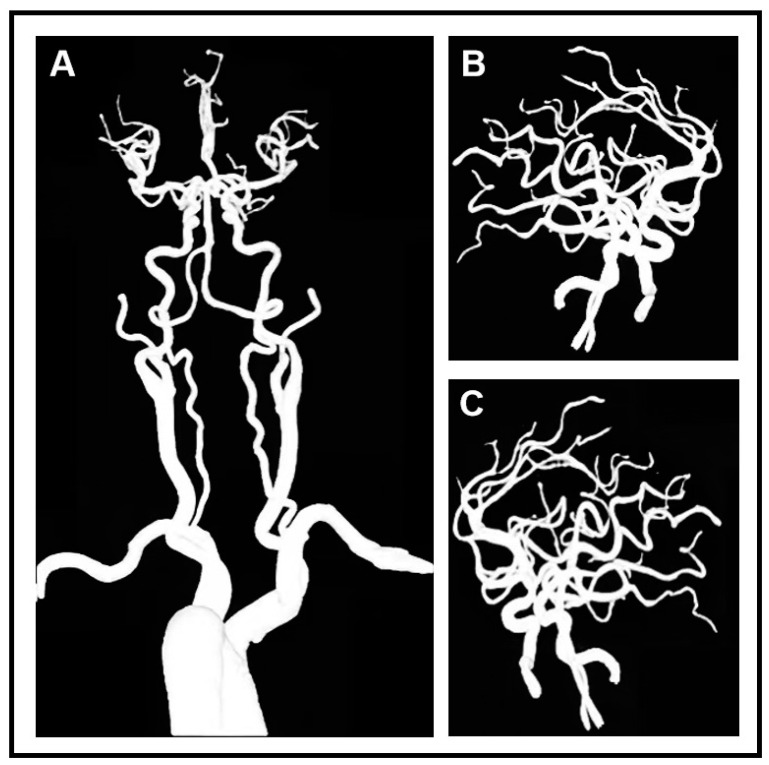
Computed tomography angiopathy (CTA) of head and neck did not reveal obvious aneurysms or abnormality in the front (**A**) and the side (**B**,**C**) of the image.

## Data Availability

No new data were created or analyzed in this study. Data sharing is not applicable to this article.

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
