# Peer review of "Mirror Image of Spontaneous Intracranial Hemorrhage"

_diagnostics, 2024, doi:10.3390/diagnostics14040357_

Round 1

Reviewer 1 Report

Comments and Suggestions for Authors

The first sentence of the second paragraph is unnecessary. If someone with a history of hypertension presents with coma there is an almost infinitesimally small chance they have this pathology.

Comments on the Quality of English Language

It is unclear what the authors mean by the "patient was insane" did they have personality changes? Agitation?

Author Response

Point-by-point response to the reviewer’s comments has upload in a Word file

Reviewer 2 Report

Comments and Suggestions for Authors

This manuscript describe a clinical case of mirror spontaneous intracranial hemorrhage. The patient had been hospitalized in a comatose state without a clear diagnosis.

There are several and important concerns about this manuscript. Firstly, the writing is confusing along text and made incompressible some paragraphs and the manuscript as a whole.

The major concern about this manuscript is the lacks of an adequate structure and information supporting the author’s arguments to be considered for publication. The authors have not adequately introduced the problem or provided a complete description of this clinical case. Furthermore, they do not provide sufficient reasons and do not discuss this case in relation to other publications to support their argues. Also, they claim that they have obtained the patient's informed consent, however, in the text they explain “the patient was insane”. Then, the authors should explain how was able to obtain such consent.

For all above, I consider this manuscript have not the quality to be publish in this journal.

Comments on the Quality of English Language

The writing is confusing and some paragraphs are not clear. It make incomprehensible the manuscript for readers 

Author Response

Point-by-point response to the reviewer’s comments has upload in a Word file.

Reviewer 3 Report

Comments and Suggestions for Authors

  1. The topic is original and relevant.
  2. An interesting clinical is described as casuistry.

Author Response

(The authors gave the same response as above.)

Round 2

Reviewer 2 Report

Comments and Suggestions for Authors

The manuscript has improved in relation to the previous version. The authors have made substantial changes following the reviewers suggestions. The manuscript now includes a more comprehensive description of the clinical case, a short discussion of important references related to this case and the authors’ hypothesis is better supported.

However, some minor changes in the manuscript should be done to improve the text and the paper comprehension

ABSTRACT. 2 sentences should by replaced:

Line 8: “This patient’s first symptom is coma” by The patient’s first symptom was the loss of consciousness…

Line 10: “This clinical presentation in conjunction with her unique imaging” by: This clinical presentation join to the atypical CT image showing bilateral hematomas indicates….

IN THE BODY OF THE MANUSCRIPT

Line 28: “Nevertheless, there are residual psychiatric disorders characterized by irritable personality alteration and behavioral abnormalities”. What this sentence means? This disorders were previously described following SSBICH by other authors? If this is the case, please, include references. Or does this sentence refer to the psychiatric disorder developed by the patient in this clinical case?

 Line 32. Acronyms such as mRS and ICH (line 39) should be specified the first time used.

So, in line 38, the sentence: “Intracranial hemorrhage is generally solitary and unilateral” should be replace by “Intracranial hemorrhage (ICH) is generally solitary and unilateral”. Likewise, in line 32, mRS must be specified

 Line 54: remove “of” from sentence: First, of bilateral…

Author Response

Dear reviewer:

On behalf of my co-authers, we thank you for your decision and constructive comments on my manuscript. We have carefully considered the suggestion of reviewer and make some changes. We study the comments carefully to improve our manuscript (diagnostics-2779002) and make revision which we hope meet with approval. All the deletions are highlightened in red text and additions are highlightened in blue text in the revised mauscript, and we upload the clean version of our mauscript at the same time. We response the comments and suggestions for authors point-by-point as follows:

Reviewer 2

1.Two sentences should be replaced in abstract.

The authers’ answer: 

In the abstract, we revised the content in Line 8 and Line 10 according to reviewer’s advise, to improve the text and the paper comprehension.

2.What the sentence Nevertheless, there are residual psychiatric disorders characterized by irritable personality alteration and behavioral abnormalities means?

The authers’ answer: 

The sentence“Nevertheless, there are residual psychiatric disorders characterized by irritable personality alteration and behavioral abnormalities” in Line 28 is aimed to provide more comprehensive description for our clinical case, rather than describe a similar characteristic following SSBICH. In order to avoid ambiguity, we add “in this patient” at the end of this sentence.

3.Acronyms such as mRS and ICH (line 39) should be specified the first time used.

The authers’ answer: 

We specify all the acronyms in our article when the first time used, including interpretation for mRS in Line 31 and ICH in Line 33.  

If there are any modifications we could make, we would like very much to modifying them. Thank you very much for you attention and time. Look forward to hearing from you.

Sincerely,

Chuanjie Wu

Department of Neurology

Xuanwu Hospital, Capital Medical University

Beijing 100053, China

Tel: +86-18911366882

Fax: 010-8315 4745

Email: wuchuanjie@ccmu.edu.cn